# Clinical characteristics and risk of second primary lung cancer after cervical cancer: A population-based study

**Chengyuan Qian[1]ʘ, Hong Liu[1]ʘ, Yan Feng[1]ʘ, Shenglan Meng[2], Dong Wang[1], Mao Nie[3]\*, Mingfang Xu**[1]\*

**1** Cancer Center, Daping Hospital, Army Medical University, Chongqing, China, **2** Department of Thoracic Surgery, Daping Hospital, Army Medical University, Chongqing, China, **3** Department of Orthopaedic Surgery, The Second Affiliated Hospital of Chongqing Medical University, Chongqing, China

ʘ These authors contributed equally to this work.
\* 302218@cqmu.edu.cm (MN); xusiyi023@126.com (MX)

**Data Availability Statement:** The data underlying the results presented in the study are available from he SEER program of the National Cancer Institute (http://seer.cancer.gov/).

## Abstract

### Background

Lung cancer is increasingly common as a second primary malignancy. However, the clinical characteristics of second primary non-small cell lung cancer after cervical cancer (CC-NSCLC) compared with first primary non-small cell lung cancer (NSCLC1) is unknown.

### Methods

The Surveillance, Epidemiology, and End Results (SEER) cancer registry between 1998 and 2010 was used to conduct a large population-based cohort analysis. The demographic and clinical characteristics, as well as prognostic data, were systematically analyzed. The overall survival (OS) in the two cohorts was further compared. The risk factors of second primary lung cancer in patients with cervical cancer were also analyzed.

### Results

A total of 557 patients (3.52%) developed second primary lung cancer after cervical cancer, and 451 were eligible for inclusion in the final analyses. Compared with NSCLC1, patients with CC-NSCLC had a higher rate of squamous cell carcinoma (SCC) (36.59% vs 19.07%, $P < 0.01$). The median OS was longer for CC-NSCLC than for NSCLC1 before propensity score matching (PSM) (16 months vs. 13 months) but with no significant difference after PSM (16 months vs. 17 months). The high-risk factors for the development of cervical cancer to CC-NSCLC include age 50–79 years, black race [odds ratio (OR) 1.417; 95% confidence interval (CI) 1.095–1.834; $P < 0.05$], and history of radiotherapy (OR 1.392; 95% CI 1.053–1.841; $P < 0.05$).

### Conclusion

Age 50–79 years, black race, and history of radiotherapy were independent risk factors for second primary lung cancer in patients with cervical cancer. Patients with CC-NSCLC had

**Funding:** This work was supported by grants from the National Natural Science Foundation of China (NSFC) No. 81502241 to CQ, and Kuanren Talents Program of the second affiliated hospital of Chongqing Medical University to MN.

**Competing interests:** The authors have declared that no competing interests exist.

distinctive clinical characteristics and better prognosis compared with patients with NSCLC1.

## Introduction

Cervical cancer is the fourth most common cancer worldwide, with an estimated 310,000 annual deaths globally [1–3]. However, improvements in early detection and cancer treatment have led to long-term survival among patients with cervical cancer. Subsequently, the possibility for patients to develop a subsequent primary cancer becomes a more important consideration [4], with a 17% higher rate of cancer in female patients compared with the general population. Moreover, cervical cancer survivors had more than double the rate of lung cancer [5]. As a matter of fact, from 1975 to 2001, 756,467 people in the United States developed a second solid cancer, representing almost 8% of the current cancer survivor population. Subsequent malignancies in cancer survivors now constitute 18% of all cancer diagnoses in the US Surveillance, Epidemiology, and End Results (SEER) cancer registries [4].

In particular, lung cancer is increasingly common as a second primary malignancy. Indeed, it is the leading cause of cancer incidence and mortality around the world, with 2.1 million lung cancer cases and 1.8 million deaths predicted in 2018, representing 18.4% cancer deaths [1]. Among women, lung cancer constitutes one of the three most commonly diagnosed cancers besides breast and colorectal cancers [2]. Notably, the incidence rates of lung cancer are now higher among young women than among young men in non-Hispanic whites and Hispanic Americans [6]. Regarding both cervical cancer and lung cancer, approximately 10% of cervical cancer survivors develop a second malignancy, of which lung cancer accounts for one of the largest numbers [7,8]. However, the risk factors for second primary lung cancer in patients with cervical cancer are not known. Similarly, differences between second primary non-small cell lung cancer after cervical cancer (CC-NSCLC) and non-small cell lung cancer (NSCLC1) in terms of clinical characteristics and survival have not been studied. Consequently, lung cancer in this specific subgroup needs to be urgently characterized in terms of its high incidence rates. Hence, this study aimed to explore the clinical differences between CC-NSCLC and NSCLC1 as well as the risk factors for second primary lung cancer in patients with cervical cancer.

## Methods

### Ethical statement

The Research Ethics Committee of Daping Hospital approved the study. Data obtained from the SEER database did not require informed patient consent because cancer is a reportable disease in the United States.

### Population

Patients with cervical cancer and NSCLC were identified from the SEER program of the National Cancer Institute (http://seer.cancer.gov/). The cohort was composed of adult patients pathologically confirmed with cervical cancer or NSCLC from the SEER database from 1998 to 2010. The accession number of the specific dataset is SEER*Stat 8.3.6. This SEER program released 18 population-based cancer registries of incidence rate and survival rate in the United States, covering about 28% of the general population. The exclusion criteria were as follows: confirmed by autopsy, unknown age of diagnosis, unknown marriage status, undetermined

grade of disease, unknown stage of disease, and unknown pathological type. Domestic statuses were recorded as follows: never married as "unmarried"; married as "married" or "unmarried but having a domestic partner"; and separated, divorced, and widowed as "other." Except for squamous cell neoplasm and adenocarcinoma, other histology types were recorded as "other," including NSCLC not otherwise specified (NSCLC-NOS).

### Statistical analysis

Categorical measurements were described as count and percentage, while continuous measurements were presented as mean (median) and range. Categorical measurements were compared using the chi-square test, while continuous measurements using the *t* test. Survival data were measured from the lung cancer date of diagnosis to the date of all-cause death or the last follow-up. Kaplan–Meier method was used to generate cumulative survival curves. Log-rank (Mantel–Cox) tests were used to compare differences in survival. Propensity score matching (PSM) was used to balance the difference in baseline characteristics between CC-NSCLC and NSCLC1 groups. According to one to three matches, 449 patients with CC-NSCLC were matched successfully. After PSM, 449 cases were found in the CC-NSCLC group and 1347 in the NSCLC1 group. No significant differences were found in histology, age at lung cancer diagnosis, race, year of lung cancer diagnosis, stage of lung cancer, marital status, radiotherapy records, chemotherapy records, surgery records, and grade between the two groups. Independent risk factors and odds ratios (OR) of second primary lung cancer were identified in patients with cervical cancer using a logistic multiple regression analysis. All *P* values were two sided, with $P < 0.05$ considered statistically significant. The incidence of second primary lung cancer was compared with previous findings. All the analyses were done using SPSS statistical software, version 23 (IBM Corp, NY, USA).

## Results

### Patient characteristics

A total of 15,809 patients with cervical cancer and 173,272 patients with NSCLC1 between 1998 and 2010 were included in this study. Of these, 557 patients (3.52%) were diagnosed with second primary non-small cell lung cancer after cervical cancer (CC-NSCLC) and 451 patients with complete information were eligible for inclusion in the final analyses. Table 1 presents the demographic and clinicopathological features of patients with NSCLC1 and CC-NSCLC. Significant differences were observed in histology, age at diagnosis, race, year at diagnosis, marital status, and cause of death between patients with CC-NSCLC and NSCLC1. No significant difference was detected in stage, radiotherapy records, chemotherapy records, surgery records, and grade. The mean time to NSCLC diagnosis was 57 months after cervical cancer, with a range of 12–192 months. The mean age at cervical cancer diagnosis was 58.2 years, whereas the mean age at CC-NSCLC diagnosis was 62.9 years. The prevalence of NSCLC was the maximum (38.36%), followed by SCC (36.39%) and other (25.1%). Of the 173,272 patients with NSCLC1 in the database, 49.05% had adenocarcinoma, 19.07% had SCC, and 31.89% were suffering from other cancer types. The proportion of SCC was apparently higher in patients with CC-NSCLC than in patients with NSCLC1 (36.59% vs 19.07%). The difference in pathologic type distribution between these two cohorts was significant ($P < 0.01$).

### Clinical features in patients with CC-NSCLC

Table 2 presents the impact of demographic characteristics and clinical features of cervical cancer on pathological types and clinical stages of lung cancer in patients with CC-NSCLC.

**Table 1.** Demographic and clinicopathological characteristics of patients with CC-NSCLC and NSCLC before and after PSM.

| Comparisons | Before PSM | | | After PSM | | |
|---|---|---|---|---|---|---|
| | NSCLC1 | CC-NSCLC | *P* value | NSCLC1 | CC-NSCLC | *P* value |
| *N* | 173,272 | 451 | | 2245 | 449 | |
| Histology | | | <0.001 | | | 0.168 |
| AC | 84,986 (49.05%) | 173 (38.36%) | | 943 (42.00%) | 173 (38.53%) | |
| SCC | 33,035 (19.07%) | 165 (36.59%) | | 714 (31.80%) | 163 (36.30%) | |
| Other | 55,251 (31.89%) | 113 (25.06%) | | 588 (26.19%) | 113 (25.17%) | |
| Age at lung cancer diagnosis (year) | | | <0.001 | | | 0.996 |
| 25–39 | 1288 (0.74%) | 13 (2.88%) | | 66 (2.94%) | 13 (2.90%) | |
| 40–49 | 9007 (5.20%) | 56 (12.42%) | | 255 (11.36%) | 55 (12.25%) | |
| 50–59 | 26,110 (15.07%) | 109 (24.17%) | | 540 (24.05%) | 109 (24.28%) | |
| 60–69 | 47,576 (27.46%) | 121 (26.83%) | | 609 (27.13%) | 121 (26.95%) | |
| 70–79 | 57,203 (33.01%) | 117 (25.94%) | | 596 (26.55%) | 116 (25.84%) | |
| ≥80 | 32,088 (18.52%) | 35 (7.76%) | | 179 (7.97%) | 35 (7.80%) | |
| Race | | | <0.001 | | | 0.47 |
| White | 145,404 (83.92%) | 338 (74.94%) | | 1662 (74.03%) | 336 (74.83%) | |
| Black | 17,839 (10.30%) | 80 (17.74%) | | 379 (16.88%) | 80 (17.82%) | |
| Other | 10,029 (5.79%) | 33 (7.32%) | | 204 (9.09%) | 33 (7.35%) | |
| Year of lung cancer diagnosis | | | <0.001 | | | 0.851 |
| 1998–2000 | 24,574 (14.18%) | 15 (3.33%) | | 65 (2.90%) | 15 (3.34%) | |
| 2001–2005 | 71,076 (41.02%) | 107 (23.73%) | | 520 (23.16%) | 106 (23.61%) | |
| 2006–2010 | 77,622 (44.80%) | 329 (72.95%) | | 1660 (73.94%) | 328 (73.05%) | |
| Stage at lung cancer diagnosis | | | 0.11 | | | 0.988 |
| Localized | 39,467 (22.78%) | 121 (26.83%) | | 597 (26.59%) | 121 (26.95%) | |
| Regional | 44,937 (25.93%) | 115 (25.50%) | | 578 (25.75%) | 115 (25.61%) | |
| Distant | 88,868 (51.29%) | 215 (47.67%) | | 1070 (47.66%) | 213 (47.44%) | |
| Marital status | | | <0.001 | | | 0.985 |
| Never married | 18,512 (10.68%) | 94 (20.84%) | | 462 (20.58%) | 93 (20.71%) | |
| Married | 73,728 (42.55%) | 185 (41.02%) | | 930 (41.43%) | 184 (40.98%) | |
| Divorced or separated or widowed | 81,032 (46.77%) | 172 (38.14%) | | 853 (38.00%) | 172 (38.31%) | |
| Radiation records | | | 0.258 | | | 0.793 |
| No | 104,795 (60.48%) | 261 (57.87%) | | 1310 (58.35%) | 259 (57.68%) | |
| Yes | 68,477 (39.52%) | 190 (42.13%) | | 935 (41.65%) | 190 (42.32%) | |
| Chemotherapy | | | 0.523 | | | 0.753 |
| No | 104,363 (60.23%) | 265 (58.76%) | | 1302 (58.00%) | 264 (58.80%) | |
| Yes | 68,909 (39.77%) | 186 (41.24%) | | 943 (42.00%) | 185 (41.20%) | |
| Surgery records | | | 0.117 | | | 0.898 |
| No | 121,139 (69.91%) | 300 (66.52%) | | 1502 (66.90%) | 299 (66.59%) | |
| Yes | 52,133 (30.09%) | 151 (33.48%) | | 743 (33.10%) | 150 (33.41%) | |
| Grade | | | 0.345 | | | 0.94 |
| I–II | 41,501 (23.95%) | 115 (25.50%) | | 587 (26.15%) | 115 (25.61%) | |
| III–IV | 52,840 (30.50%) | 146 (32.37%) | | 702 (31.27%) | 144 (32.07%) | |
| Unknown | 78,931 (45.55%) | 190 (42.13%) | | 956 (42.58%) | 190 (42.32%) | |
| Cause of death or alive | | | <0.001 | | | <0.001 |
| Lung cancer | 120,106 (69.32%) | 202 (44.79%) | | 1481 (65.97%) | 201 (44.77%) | |
| Cervical cancer | 113 (0.07%) | 53 (11.75%) | | 12 (0.53%) | 52 (11.58%) | |
| Alive | 18,763 (10.83%) | 115 (25.50%) | | 377 (16.79%) | 115 (25.61%) | |
| Other | 34,290 (19.79%) | 81 (17.96%) | | 375 (16.70%) | 81 (18.04%) | |

**Table 2. Impact of demographic and clinical features of cervical cancer on pathological types and clinical stages of lung cancer.**

| Cervical cancers | Lung cancers | | | | P value | Lung cancers | | | P value |
|---|---|---|---|---|---|---|---|---|---|
| | Total | AC (%) | SCC (%) | Other (%) | | Localized (%) | Regional (%) | Distant (%) | |
| Total | 451 | 173 (38.4) | 165 (36.6) | 113 (25.1) | | 121 (26.8) | 115 (25.5) | 215 (47.7) | |
| Latency | | | | | <0.05 | | | | >0.05 |
| ≤1 year | 100 | 42 (42.0) | 31 (31.0) | 27 (27.0) | | 37 (37.0) | 23 (23.0) | 40 (40.0) | |
| >1 year, ≤5 years | 187 | 57 (30.5) | 69 (36.9) | 61 (32.6) | | 43 (23.0) | 48 (25.7) | 96 (51.3) | |
| >5 years,≤10 years | 117 | 51 (43.6) | 47 (40.2) | 19 (16.2) | | 27 (23.1) | 34 (29.0) | 56 (47.9) | |
| >10 years | 47 | 23 (48.9) | 18 (38.3) | 6 (12.8) | | 14 (29.8) | 10 (21.3) | 23 (48.9) | |
| Race | | | | | >0.05 | | | | >0.05 |
| White | 338 | 127 (37.6) | 122 (36.1) | 89 (26.3) | | 97 (28.7) | 84 (24.9) | 157 (46.4) | |
| Black | 80 | 29 (36.3) | 36 (45.0) | 15 (18.8) | | 16 (20.0) | 23 (28.7) | 41 (51.3) | |
| Other | 33 | 17 (51.5) | 7 (21.2) | 9 (27.3) | | 8 (24.2) | 8 (24.2) | 17 (51.6) | |
| Age at cervical cancer diagnosis (year) | | | | | >0.05 | | | | >0.05 |
| 25–39 | 35 | 13 (37.1) | 15 (42.9) | 7 (20.2) | | 8 (22.9) | 10 (28.6) | 17 (48.6) | |
| 40–49 | 77 | 20 (26.0) | 33 (42.9) | 24 (31.1) | | 16 (20.8) | 19 (24.7) | 42 (54.5) | |
| 50–59 | 133 | 61 (45.9) | 39 (29.3) | 33 (24.8) | | 32 (24.1) | 35 (26.3) | 66 (49.6) | |
| 60–69 | 124 | 53 (42.7) | 43 (34.7) | 28 (22.6) | | 39 (31.5) | 33 (26.6) | 52 (41.9) | |
| 70–79 | 68 | 21 (30.9) | 28 (41.2) | 19 (27.9) | | 20 (29.4) | 17 (25.0) | 31 (45.6) | |
| ≥80 | 14 | 5 (35.7) | 7 (50.5) | 2 (14.3) | | 6 (42.9) | 1 (7.1) | 7 (50.0) | |
| Year of cervical cancer diagnosis | | | | | >0.05 | | | | >0.05 |
| 1998–2000 | 112 | 36 (32.1) | 43 (38.4) | 33 (29.5) | | 28 (25.0) | 30 (26.8) | 54 (48.2) | |
| 2001–2005 | 216 | 84 (38.9) | 81 (37.5) | 51 (23.6) | | 62 (28.7) | 51 (23.6) | 103 (47.7) | |
| 2006–2010 | 123 | 53 (43.1) | 41 (33.3) | 29 (23.6) | | 31 (25.2) | 34 (27.6) | 58 (47.2) | |
| Stage at CC diagnosis | | | | | <0.001 | | | | >0.05 |
| Localized | 213 | 103 (48.4) | 61 (28.6) | 49 (23.0) | | 60 (28.2) | 57 (26.8) | 96 (45.0) | |
| Regional | 238 | 70 (29.4) | 104 (43.7) | 64 (26.9) | | 61 (25.6) | 58 (24.4) | 119 (50.0) | |
| Histology | | | | | p<0.001 | | | | >0.05 |
| AC | 89 | 49 (55.0) | 15 (16.9) | 25 (28.1) | | 20 (22.5) | 24 (27.0) | 45 (50.5) | |
| SCC | 335 | 111 (33.1) | 146 (43.6) | 78 (23.3) | | 94 (28.1) | 85 (25.4) | 156 (46.6) | |
| Other | 27 | 13 (48.2) | 4 (14.8) | 10 (37.0) | | 7 (25.9) | 6 (22.2) | 14 (51.9) | |
| Marital status | | | | | >0.05 | | | | >0.05 |
| Never married | 94 | 33 (35.1) | 38 (40.4) | 23 (24.5) | | 19 (20.2) | 24 (25.5) | 51 (54.3) | |
| Married | 185 | 82 (44.3) | 57 (30.8) | 46 (24.9) | | 50 (27.0) | 48 (25.9) | 87 (47.1) | |
| Divorced or separated or widowed | 172 | 58 (33.7) | 70 (40.7) | 44 (25.6) | | 52 (30.2) | 43 (25.0) | 77 (44.8) | |
| Radiation records | | | | | <0.001 | | | | >0.05 |
| No | 151 | 78 (51.7) | 37 (24.5) | 36 (23.8) | | 82 (27.3) | 70 (23.3) | 148 (49.4) | |
| Yes | 300 | 95 (31.7) | 128 (42.7) | 77 (25.6) | | 39 (25.8) | 45 (29.8) | 67 (44.4) | |
| Chemotherapy records | | | | | <0.001 | | | | >0.05 |
| No | 247 | 117 (47.4) | 74 (30.0) | 56 (22.6) | | 55 (27.0) | 46 (22.5) | 103 (50.5) | |
| Yes | 204 | 56 (27.5) | 91 (44.6) | 57 (27.9) | | 66 (26.7) | 69 (27.9) | 112 (45.3) | |
| Surgery records | | | | | | | | | >0.05 |
| No | 194 | 56 (28.9) | 87 (44.8) | 51 (26.3) | | 69 (26.8) | 63 (24.5) | 125 (48.7) | |
| Yes | 257 | 117 (45.5) | 78 (30.4) | 62 (24.1) | | 52 (26.8) | 52 (26.8) | 90 (46.4) | |
| Grade | | | | | <0.05 | | | | >0.05 |
| I–II | 180 | 77 (42.8) | 69 (38.3) | 34 (18.9) | | 55 (30.6) | 48 (26.7) | 77 (42.8) | |
| III–IV | 162 | 51 (31.5) | 64 (39.5) | 47 (29.0) | | 33 (20.4) | 37 (22.8) | 92 (56.8) | |
| Unknown | 109 | 45 (41.3) | 32 (29.4) | 32 (29.4) | | 33 (30.3) | 30 (27.5) | 46 (42.2) | |

Latency, stage, histology, radiotherapy records, chemotherapy records, and grade of cervical cancer were associated with pathological types of lung cancer, rather than race, age at cervical cancer diagnosis, year of cervical cancer diagnosis, marital status, and surgery records. No significant correlation was found between the clinical factors of cervical cancer and the stages of lung cancer in patients with CC-NSCLC.

Table 3 presents the impact of demographic characteristics and clinical features of cervical cancer on the cause of death in patients with CC-NSCLC. Latency, age at diagnosis, stage, histology, marital status, radiotherapy records, chemotherapy records, surgery records, and grade were associated with the causes of death, rather than race and year of cervical cancer diagnosis. Patients with a latency of ≤1year were more likely to die of cervical cancer, and those with a latency of >5 years were more likely to survive. Married patients with young age, regional stage, and treated with radiation or chemotherapy died more often from cervical cancer. Patients with cervical adenocarcinoma, well or moderately differentiated in terms of histological grade and treated with surgery, were more likely to survive. Lung cancer was the most common cause of death (44.8%).

## Risk factors of second primary lung cancer in patients with cervical cancer

High-risk factors of developing second primary lung cancer in patients with cervical cancer include age between 50 and 79 years, black race, and history of radiotherapy. Table 4 presents all significant independent factors for the development of second primary lung cancer, obtained from the logistic multiple regression analysis.

## Survival analysis

The median OS was 16 months (range, 1–191 months) versus 13 months (range, 1–227 months) in patients with CC-NSCLC and patients with NSCLC1 before PSM, respectively, but 16 months versus 17 months after PSM, respectively. The difference was significant before PSM ($P < 0.05$) but not significant after PSM ($P > 0.05$). Fig 1 shows the survival curves of CC-NSCLC and NSCLC1 before and after PSM. OS was longer for CC-NSCLC versus NSCLC1 before PSM but showed no significant difference after PSM.

The stage was strongly associated with OS. The median OS of localized, regional, and distant CC-NSCLC was 52.0, 25.0, and 8.0 months, respectively ($P < 0.0001$). The pathological type was another important prognostic factor for OS. Patients with adenocarcinoma had much longer OS compared with those with SCC or other [22.0, 16.0, and 11.0 months, respectively ($P < 0.01$)]. Young patients had superior OS compared with patients older than 80 years. The latter had a median OS of 7 months. Differentiation was also an important prognostic factor. The mOS of patients with CC-NSCLC having grade I + II, grade III + IV, and unknown differentiation was 45, 13, and 10 months, respectively ($P < 0.01$). Patients who had surgery had a better prognosis (70 months vs 10 months, $P < 0.01$, and those who were exposed to radiation had a worse prognosis (13 months vs 121 months, $P < 0.01$) (Fig 2).

## Discussion

Over the past three decades, advances in the early detection and treatment of cervical cancer have resulted in a significant improvement in survival among patients with cervical cancer. Survival after a cervical cancer diagnosis is higher nowadays due to improvements in cancer therapy; however, an emerging issue in survivors is the occurrence of second cancer [9]. Patients develop subsequent primary cancers as a result of shared lifestyle and genetic factors, as well as the first cancer treatment. In particular, lung cancer accounts for one of the largest numbers in cervical cancer survivors who have developed a second malignancy [7,8].

**Table 3. Impact of demographic and clinical features of cervical cancer on death in patients with CC-NSCLC.**

| Cervical cancers | Total | Cause of death or alive | | | | P value |
|---|---|---|---|---|---|---|
| | | Lung and bronchus (%) | Cervix uteri (%) | Alive (%) | Other (%) | |
| Total | 451 | 202 (44.8) | 53 (11.8) | 115 (25.5) | 81 (18.0) | |
| Latency | | | | | | <0.001 |
| ≤1 year | 100 | 42 (42.0) | 18 (18.0) | 12 (12.0) | 28 (28.0) | |
| >1 year, ≤5 years | 187 | 86 (46.0) | 24 (12.8) | 42 (22.5) | 35 (18.7) | |
| >5 years, ≤10 years | 117 | 54 (46.2) | 9 (7.7) | 40 (34.2) | 14 (12.0) | |
| >10 years | 47 | 20 (42.6) | 2 (4.3) | 21 (44.7) | 4 (8.5) | |
| Race | | | | | | >0.05 |
| White | 338 | 153 (45.3) | 43 (12.7) | 85 (25.1) | 57 (16.9) | |
| Black | 80 | 35 (43.8) | 8 (10.0) | 18 (22.5) | 19 (23.8) | |
| Other | 33 | 14 (42.4) | 2 (6.1) | 12 (36.4) | 5 (15.2) | |
| Age at cervical cancer diagnosis (year) | | | | | | <0.001 |
| 25–39 | 35 | 7 (20.0) | 10 (28.6) | 15 (42.9) | 3 (8.5) | |
| 40–49 | 77 | 35 (45.5) | 11 (14.3) | 19 (24.7) | 12 (15.5) | |
| 50–59 | 133 | 61 (45.9) | 16 (12.0) | 38 (28.6) | 18 (13.5) | |
| 60–69 | 124 | 55 (44.4) | 11 (8.9) | 33 (26.6) | 25 (20.1) | |
| 70–79 | 68 | 37 (54.4) | 3 (4.4) | 8 (11.8) | 20 (29.4) | |
| ≥80 | 14 | 7 (50.0) | 2 (14.3) | 2 (14.3) | 3 (21.4) | |
| Year of cervical cancer diagnosis | | | | | | >0.05 |
| 1998–2000 | 112 | 51 (45.5) | 14 (12.5) | 26 (23.2) | 21 (18.8) | |
| 2001–2005 | 216 | 104 (48.1) | 27 (12.5) | 55 (25.5) | 30 (13.9) | |
| 2006–2010 | 123 | 47 (38.2) | 12 (9.8) | 34 (27.6) | 30 (24.4) | |
| Stage at breast cancer diagnosis | | | | | | <0.05 |
| Localized | 213 | 96 (45.1) | 12 (5.6) | 67 (31.5) | 38 (17.8) | |
| Regional | 238 | 106 (44.5) | 41 (17.2) | 48 (20.2) | 43 (18.1) | |
| Histology | | | | | | p<0.05 |
| AC | 89 | 32 (36.0) | 7 (7.9) | 33 (37.1) | 17 (19.0) | |
| SCC | 335 | 155 (46.3) | 41 (12.2) | 80 (23.9) | 59 (17.6) | |
| Other | 27 | 15 (55.6) | 5 (18.5) | 2 (7.4) | 5 (18.5) | |
| Marital status | | | | | | <0.05 |
| Never married | 94 | 44 (46.8) | 7 (7.4) | 26 (27.7) | 17 (18.1) | |
| Married | 185 | 75 (40.5) | 29 (15.7) | 56 (30.3) | 25 (13.5) | |
| Divorced or separated or widowed | 172 | 83 (48.3) | 17 (9.9) | 33 (19.2) | 39 (22.7) | |
| Radiation records | | | | | | <0.05 |
| Yes | 300 | 134 (44.7) | 46 (15.3) | 64 (21.3) | 56 (18.7) | |
| No | 151 | 68 (45.0) | 7 (4.6) | 51 (33.8) | 25 (16.6) | |
| Chemotherapy records | | | | | | <0.05 |
| Yes | 204 | 85 (41.7) | 37 (18.1) | 47 (23.0) | 35 (17.2) | |
| No | 247 | 117 (47.4) | 16 (6.5) | 68 (27.5) | 46 (18.6) | |
| Surgery records | | | | | | <0.05 |
| Yes | 257 | 110 (42.8) | 24 (9.3) | 83 (32.3) | 40 (15.6) | |
| No | 194 | 92 (47.4) | 29 (14.9) | 32 (16.5) | 41 (21.1) | |
| Grade | | | | | | <0.05 |
| I–II | 180 | 73 (40.6) | 18 (10.0) | 60 (33.3) | 29 (16.1) | |
| III–IV | 162 | 82 (50.6) | 24 (14.8) | 32 (19.8) | 24 (14.8) | |
| Unknown | 109 | 47 (43.1) | 11 (10.1) | 23 (21.1) | 28 (25.7) | |

**Table 4. Risk factors for second primary lung cancer in patients with cervical cancer.**

| Comparisons | Cervical cancer | CC-NSCLC | OR (95% CI) | P value |
|---|---|---|---|---|
| Histology | | | | >0.05 |
| AC | 3,705 | 89 (19.73%) | | |
| SCC | 10,506 | 335 (74.28%) | 1.137 (0.890–1.454) | |
| Other | 1,147 | 27 (5.99%) | 0.832 (0.533–1.297) | |
| Age (year) | | | | <0.05 |
| 25–39 | 5,101 (33.21%) | 35 (7.76%) | | |
| 40–49 | 4,413 (28.73%) | 77 (17.07%) | 2.410 (1.609–3.609) | |
| 50–59 | 2,680 (17.45%) | 133 (29.49%) | 6.458 (4.398–9.483) | |
| 60–69 | 1,588 (10.34%) | 124 (27.49%) | 9.956 (6.712–14.767) | |
| 70–79 | 993 (6.47%) | 68 (15.08%) | 8.464 (5.470–13.099) | |
| ≥80 | 583 (3.80%) | 14 (3.10%) | 2.761 (1.427–5.342) | |
| Race | | | | <0.05 |
| White | 11,492 (74.83%) | 338 (74.94%) | | |
| Black | 1,716 (11.17%) | 80 (17.74%) | 1.417 (1.095–1.834) | |
| Other | 2,150 (14.00%) | 33 (7.32%) | 0.432 (0.300–0.622) | |
| Year | | | | >0.05 |
| 1998–2000 | 4,008 (26.10%) | 112 (24.83%) | | |
| 2001–2005 | 6,224 (40.53%) | 216 (47.89%) | 1.177 (0.929–1.492) | |
| 2006–2010 | 5,126 (33.38%) | 123 (27.27%) | 0.819 (0.627–1.069) | |
| Stage | | | | >0.05 |
| Localized | 9,123 (59.40%) | 213 (47.23%) | | |
| Regional | 6,235 (40.60%) | 238 (52.77%) | 0.877 (0.684–1.124) | |
| Marital status | | | | >0.05 |
| Unmarried | 4,480 (29.17%) | 94 (20.84%) | | |
| Married | 7,371 (47.99%) | 185 (41.02%) | 1.165 (0.898–1.510) | |
| Divorced or separated or widowed | 3,507 (22.84%) | 172 (38.14%) | 1.391 (1.062–1.822) | |
| Surgery records | | | | >0.05 |
| No | 4,687 (30.52%) | 194 (43.02%) | | |
| Yes | 10,671 (69.48%) | 257 (56.98%) | 0.998 (0.768–1.267) | |
| Radiation records | | | | <0.05 |
| No | 7,820 (50.92%) | 151 (33.48%) | | |
| Yes | 7,538 (49.08%) | 300 (66.52%) | 1.392 (1.053–1.841) | |
| Chemotherapy records | | | | >0.05 |
| No | 9,904 (64.49%) | 247 (54.77%) | | |
| Yes | 5,454 (35.51%) | 204 (45.23%) | 0.989 (0.764–1.280) | |
| Grade | | | | >0.05 |
| I-II | 6,100 (39.72%) | 180 (39.91%) | | |
| III–IV | 4,485 (29.20%) | 162 (35.92%) | 1.057 (0.847–1.319) | |
| Unknown | 4,773 (31.08%) | 109 (24.17%) | 0.814 (0.634–1.045) | |

However, the risk factors for second primary lung cancer in patients with cervical cancer are poorly known. Similarly, whether the clinical features of cervical cancer have an impact on pathological types and stages of second lung cancer is not known. Furthermore, survival data are lacking in terms of differences in prognosis between NSCLC1 and CC-NSCLC. This study was novel in focusing on CC-NSCLC, with the aim to explore objective differences between CC-NSCLC and NSCLC1 besides the goal to identify risk factors for second primary lung cancer in patients with cervical cancer.

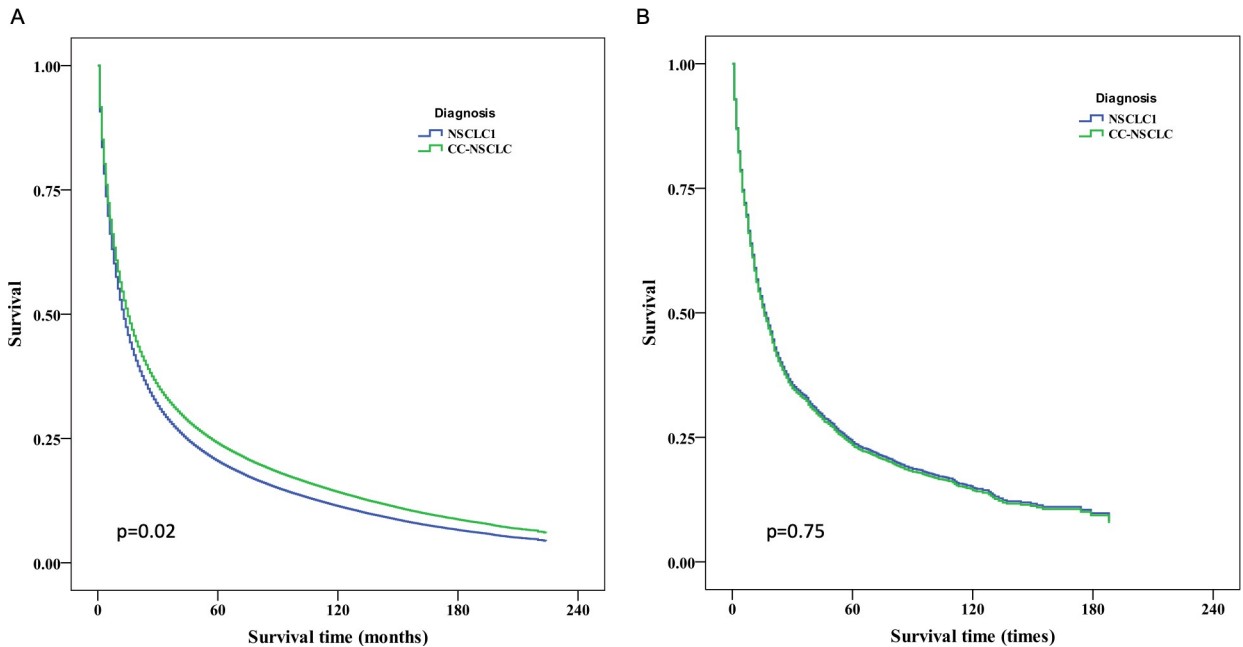

**Fig 1. mOS for CC-NSCLC and NSCLC1 before and after PSM.** (A) Kaplan–Meier survival curves indicated that patients with CC-NSCLC had significantly longer survival compared with patients with NSCLC1 before PSM (16 months vs 13 months; *P* < 0.05). (B) Kaplan–Meier survival curves indicated that patients with CC-NSCLC had no significant extension in survival relative to patients with NSCLC1 after PSM (16 months vs 15 months; *P* > 0.05).

The study provided answers to the following issues. First and foremost, patients with CC-NSCLC were younger with earlier stages. The proportion of SCC was significantly higher in patients with CC-NSCLC than in patients with NSCLC1 (36.59% vs 19.07%). If patients with CC-NSCLC seemed to have a better prognosis, no significant difference was found after PSM. Concerning epidemiologic data, the incidence of lung cancer differs according to the geographical region and over time. In particular, both incidence and mortality from lung cancer continue to increase sharply in China [10,11]. The results of this study suggested that the incidence of lung cancer among cervical cancer survivors in the present cohort (3.52%) was significantly higher compared with the rates reported in the literature. According to the Region-Specific Incidence Age-Standardized Rates by Sex for Cancers of the Lung in 2018, women in Northern America had the highest incidence, which was 30.7 per 100,000 [1].

In the present cohort, patients with CC-NSCLC were younger than patients with NSCLC1 and displayed earlier stages, which might be due to more frequent medical examinations. Squamous cell lung carcinoma was the most common histologic subtype before the 1990s. Currently, adenocarcinoma has become the most common histologic subtype of lung cancer in men and women [12,13]. Although adenocarcinoma (38.36%) remains the most common histologic subtype in patients with CC-NSCLC, the proportion of SCC is significantly higher among patients with NSCLC1 (36.59% vs 19.07%). SCC accounted for 19.07% of patients with NSCLC1 in the present study, similar to data in the literature [14]. The high proportion of SCC in patients with cervical cancer might be due to the history of chemo-radiation therapy as well as confounding factors such as unclear primary cancer and cervical cancer metastasis. Besides, since 2006, the incidence of second primary lung cancer among patients with cervical cancer drastically increased in the present study, which could not be entirely explained by the increase in lung cancer rates. Indeed, causes for the increased incidence of second primary lung cancer in patients with cervical cancer need further exploration. No difference was

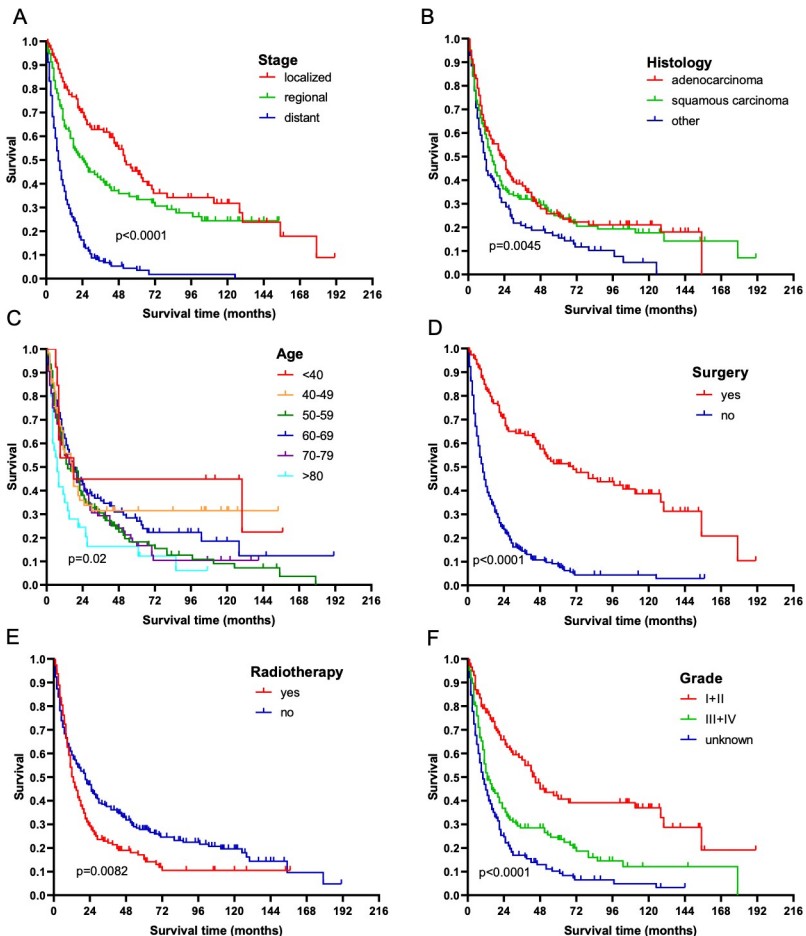

**Fig 2. Influence of stage, pathological type, age, surgical records, radiotherapy records, and differentiation grade on OS in CC-NSCLC.** (A) Kaplan–Meier survival curves indicated that the mOS of patients with CC-NSCLC having localized, regional, and distant staging was 52.0, 25.0, and 8.0 months, respectively ($P < 0.0001$). (B) Kaplan–Meier survival curves indicated that the mOS of patients with CC-NSCLC having adenocarcinoma, SCC, and other was 22.0, 16.0, and 11.0 months, respectively ($P < 0.01$). Kaplan–Meier survival curves indicated the following: (C) young patients with CC-NSCLC had superior OS compared with patients older than 80 years ($P < 0.05$); (D) patients with CC-NSCLC who underwent surgical resection had significantly longer survival compared with those who did not undergo resection (70 months vs 10 months; $P < 0.0001$); (E) patients with CC-NSCLC who underwent radiotherapy had significantly shorter survival compared with those who did not undergo radiotherapy (13 months vs 21 months; $P < 0.01$); and (F) the mOS of patients with CC-NSCLC having grade I + II, grade III + IV, and unknown differentiation was 45, 13, and 10 months, respectively ($P < 0.0001$).

observed in the number of patients undergoing surgery, chemotherapy. and radiotherapy between the two groups. The cancer-related death rate was significantly lower and the survival rate was significantly higher (25.50% vs 10.83%) in patients with CC-NSCLC than in patients with NSCLC1 (56.54% vs 69.39%), with longer OS in patients with CC-NSCLC. However, no significant difference was found in prognosis after PSM between CC-NSCLC and NSCLC1. There is no doubt that tumor staging is the most important factor in survival prognostication. In addition to tumor staging, other factors that have been shown to have prognostic value include grade of tumor, sex, age, smoking status, general condition, complications, operation type, etc. [15,16] Since there was no significant difference in staging before PSM, we inferred that the better prognosis of CC-NSCLC before PSM might be due to being younger of patients. Therefore, we believe that there is no significant difference in prognosis between CC-NSCLC

and NSCLC1 if there is no significant difference in patient's general condition, pathological and stage of tumor. Patients with surgical records but without radiotherapy records had a significantly better prognosis. This phenomenon can be partly explained by the fact that more patients in the early stage received surgical treatment and more patients in the late stage received radiotherapy. Therefore, patients with cervical cancer having high-risk factors, including age 50–79 years, black race, and history of radiotherapy, should be reexamined with chest computed tomography more frequently for the early diagnosis of second primary lung cancer.

Regarding the impact of clinical features of patients with cervical cancer on the pathological types of lung cancer, the present study highlighted that the incidence of SCC was higher in blacks than in whites among patients with CC-NSCLC (45.0% vs 36.1%), which was consistent with previous findings [17]. Lung squamous cancer is more common in patients with longer latency, regional stage, and history of cervical SCC. A previous study reported that the second lung SCC was more common in patients with cervical SCC compared with AC survivors, whereas the second lung AC was more common in patients with cervical AC compared with SCC survivors [18]. Thus, some patients diagnosed with lung SCC may be metastatic from cervical cancer. Besides, patients with a history of chemo-radiation therapy also had a higher rate of SCC. A study [19] showed that radiation penetrated the epidermis sufficiently to induce irreversible DNA damage in cells located beneath the epidermis, causing SCC. Cervical cancer variables did not affect the stage in patients with CC-NSCLC.

A population-based study showed that age, chemotherapy, and radiotherapy were independent risk factors for the second primary malignancy among cancer survivors [20]. However, standardized incidence ratios were not significantly higher in the radiation group in another large population-based study using SEER data. Half of the patients had none/unknown status of radiotherapy, explaining the results [21]. In the present population-based study using SEER data, significantly high incidence ratios were observed in the radiation group. The application of intensity-modulated radiation therapy (IMRT) and three-dimensional conformal radiation therapy (3D-CRT) has become increasingly common with the improvement in radiotherapy technology [22,23]. The present study could not assess whether IMRT and 3D-CRT increased the risk of a second primary tumor.

Finally, the present study had several limitations, including the lack of biological data on the PD-L1 (programmed cell death-1) status [24–26]. This crucial data could not be collected because biomarker analysis was not universalized in clinical practice until very recent years. Consequently, further studies are warranted for targetable driver mutations and PD-L1 in CC-NSCLC. Furthermore, this study did not include the influence of interactions of all possible risk factors on patients with cervical cancer. Additionally, the main limitation of the SEER data, like any retrospective study on treatment effects, is the lack of randomness in treatment regimens, leading to confounding factors. Hence, the result may be biased and should be interpreted with caution despite the use of PSM to remove this defect.

## Author Contributions

**Conceptualization:** Dong Wang, Mingfang Xu.

**Data curation:** Shenglan Meng, Mingfang Xu.

**Formal analysis:** Yan Feng, Mingfang Xu.

**Funding acquisition:** Chengyuan Qian.

**Investigation:** Yan Feng, Mao Nie.

**Methodology:** Chengyuan Qian, Mao Nie.

**Project administration:** Chengyuan Qian, Hong Liu.

**Supervision:** Hong Liu, Yan Feng.

**Writing – original draft:** Chengyuan Qian.

**Writing – review & editing:** Dong Wang, Mao Nie, Mingfang Xu.

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
