## [Decision Letter · Decision Letter 0]

29 May 2020

PONE-D-20-09128

Clinical Characteristics and Risk of Second Primary Lung Cancer After Cervix Cancer: A Population-Based Study

PLOS ONE

Dear Dr. Xu,

Thank you for submitting your manuscript to PLOS ONE. After careful consideration, we feel that it has merit but does not fully meet PLOS ONE’s publication criteria as it currently stands. Therefore, we invite you to submit a revised version of the manuscript that addresses the points raised during the review process.

We look forward to receiving your revised manuscript.

Kind regards,

Jean-Louis Pujol, M.D.

Academic Editor

PLOS ONE

Journal Requirements:

3. In the Methods section, please provide the accession number of the specific dataset downloaded from the SEER database for this study.

Reviewers' comments:

Reviewer's Responses to Questions

**Comments to the Author**

1. Is the manuscript technically sound, and do the data support the conclusions?

Reviewer #1: Yes

2. Has the statistical analysis been performed appropriately and rigorously? 

Reviewer #1: Yes

3. Have the authors made all data underlying the findings in their manuscript fully available?

Reviewer #1: Yes

4. Is the manuscript presented in an intelligible fashion and written in standard English?

Reviewer #1: Yes

5. Review Comments to the Author

Reviewer #1: This study focus on clinical differences between CC-NSCLC and NSCLC1 as well as risk factors for secondary primary lung cancer in patients with cervical cancer

Major comment

In fact this study is a negative study because after propensity score matching, there is no difference. This is the main results and should be reported is the abstract and highlight in the discussion.

Discussion should be more concise in my opinion as in general population age, black race and radiotherapy are well know risk factors for lung cancer

Minor comments

Methods section, population paragraph, the number of patients is reported on the results section, the delete from this section

Some references are missed ie .J Clin Oncol. 2009 Feb 20;27(6):967-73. Second cancers after squamous cell carcinoma and adenocarcinoma of the cervix. Chaturvedi AK1, Kleinerman RA, Hildesheim A, Gilbert ES, Storm H, Lynch CF, Hall P, Langmark F, Pukkala E, Kaijser M, Andersson M, Fossa SD, Joensuu H, Travis LB, Engels EA.

Secondary Primary Malignancy Risk in Patients With Cervical Cancer in Taiwan: A Nationwide Population-Based Study.

Teng CJ, Huon LK, Hu YW, Yeh CM, Chao Y, Yang MH, Chen TJ, Hung YP, Liu CJ. Medicine (Baltimore). 2015 Oct;94(43)

6. PLOS authors have the option to publish the peer review history of their article (what does this mean?). If published, this will include your full peer review and any attached files.

Reviewer #1: Yes: christos chouaid

---

## [Author Response · Author response to Decision Letter 0]

23 Jun 2020

RE: Manuscript ID # PONE-D-20-09128

Dear Editor,

We would like to thank PLOS ONE for giving us the opportunity to revise our manuscript. 

We thank the reviewers and editors for their careful read and thoughtful comments on previous draft. We have carefully taken their comments into consideration in preparing our revision, which has resulted in a paper that is clearer and more compelling.

Below is our point-by-point response to the reviewer#1’s comments.

We want to begin by thanking Reviewer #1 for positive comments . We addressed all the points raised by the reviewer, as summarized below.

1. According to the major comment ,as a negative study mentioned by the reviewer ,we have emphasized this result in the abstract（line 37 in Revised Manuscript with Track Changes） and highlighted it in the discussion section（line 267-275 in Revised Manuscript with Track Changes）.

2. As suggested by the reviewer,in order to make the discussion more concise,we have deleted discussion about age, black race and radiotherapy as risk factors for lung cancer in discussion(line 236-238 in Revised Manuscript with Track Changes)

3. According to the minor comment , as suggested by the reviewer, we have deleted the number of patients in methods section, population paragraph(line 91-92 in Revised Manuscript with Track Changes) 

4. As suggested by the reviewer, we have added two references18,20(line288-290,line296-297 in Revised Manuscript with Track Changes).

Below is our point-by-point response to the editor’s comments and additional requirements,including：

1. we have revised the manuscript to meet the PLOS ONE's style requirements.

2. We have thoroughly copyedited our manuscript for language usage, spelling, and grammar with help of a professional scientific editing service named Mjeditor (www.mjeditor.com). A copy of my manuscript showing changes by using track changes and an editorial certificate uploaded as *supporting information* files.

3. In the Methods section, we added the accession number of the specific dataset downloaded from the SEER database for this study（line 86-87 in Revised Manuscript with Track Changes）.

4. I have authenticated a pre-existing ORCID ID in Editorial Manager.

5. We have made changes to our financial disclosure and included the updated statement in our cover letter.

Thanks for all the help.

Best wishes,

Dr Mingfang Xu

Corresponding Author

2020-06-22

---

## [Decision Letter · Decision Letter 1]

15 Jul 2020

Clinical Characteristics and Risk of Second Primary Lung Cancer After Cervix Cancer: A Population-Based Study

PONE-D-20-09128R1

Dear Dr. Xu,

We’re pleased to inform you that your manuscript has been judged scientifically suitable for publication and will be formally accepted for publication once it meets all outstanding technical requirements.

Kind regards,

Jean-Louis Pujol, M.D.

Academic Editor

PLOS ONE

Additional Editor Comments (optional):

Reviewers' comments:

Reviewer's Responses to Questions

**Comments to the Author**

1. If the authors have adequately addressed your comments raised in a previous round of review and you feel that this manuscript is now acceptable for publication, you may indicate that here to bypass the “Comments to the Author” section, enter your conflict of interest statement in the “Confidential to Editor” section, and submit your "Accept" recommendation.

Reviewer #1: All comments have been addressed

2. Is the manuscript technically sound, and do the data support the conclusions?

Reviewer #1: Yes

3. Has the statistical analysis been performed appropriately and rigorously? 

Reviewer #1: Yes

4. Have the authors made all data underlying the findings in their manuscript fully available?

Reviewer #1: Yes

5. Is the manuscript presented in an intelligible fashion and written in standard English?

Reviewer #1: Yes

6. Review Comments to the Author

Reviewer #1: no more comments, authors respond to all the questions and clarify the manuscript, references is adequate

7. PLOS authors have the option to publish the peer review history of their article (what does this mean?). If published, this will include your full peer review and any attached files.

Reviewer #1: **Yes: **christos chouaid

---

## [Editor Report · Acceptance letter]

23 Jul 2020

PONE-D-20-09128R1 

Clinical characteristics and risk of second primary lung cancer after cervical cancer: A population-based study 

Dear Dr. Xu:

I'm pleased to inform you that your manuscript has been deemed suitable for publication in PLOS ONE. Congratulations! Your manuscript is now with our production department. 

Kind regards, 

on behalf of

Dr. Jean-Louis Pujol 

Academic Editor

PLOS ONE